# Measuring Stereotypes in Interprofessional Education: A Pilot High-Fidelity Simulation Study Among Postgraduate Nursing and Physician Students in a Spanish University

**DOI:** 10.3390/healthcare12232449

**Published:** 2024-12-05

**Authors:** Juan Manuel Cánovas-Pallarés, Sergio Nieto-Caballero, Manuel Baeza-Mirete, Manuel José Párraga-Ramírez, Andrés Rojo-Rojo

**Affiliations:** 1Emergency Healthcare System, SAMU, Public Valencian Health Service, SAMU-Alicante, Pintor Baeza, nº 12, 03010 Alicante, Spain; jmcanovas@ucam.edu; 2Faculty of Nursing, Catholic University of Murcia (UCAM), Av. de los Jerónimos, nº 135, 30107 Murcia, Spain; 3Emergency Healthcare System, Public Murcian Health Service, Escultor José Sánchez Lozano, 7, 2ª Planta, 30005 Murcia, Spain; 4Faculty of Medicine, Catholic University of Murcia (UCAM), Av. de los Jerónimos, nº 135, 30107 Murcia, Spain; 5Intensive Care Unit, Virgen de la Arrixaca Universitary Hospital, Public Murcian Health Service, Ctra. Madrid-Cartagena, s/n, 30120 Murcia, Spain; 6Intensive Care Unit, Morales Meseguer Universitary Hospital, Public Murcian Health Service, Av. Marqués de Los Vélez, s/n, 30008 Murcia, Spain

**Keywords:** nursing stereotypes, interprofessional education (IPE), high-fidelity simulation, postgraduate students, teamwork collaboration

## Abstract

Background/Objectives: Nursing professionals are often subject to social stereotypes that can hinder effective teamwork with other healthcare professionals and limit their professional growth. Interprofessional education (IPE) enhances teamwork skills and promotes a better understanding of other professional groups. This study aimed to identify the presence of stereotypes associated with nursing among postgraduate nursing and student physicians specializing in emergency medicine and to assess the applicability of simulation as an IPE strategy. Methods: A pilot study using high-fidelity simulation activity focusing on interdisciplinary collaboration was designed for students in the master’s programs in emergency nursing and emergency medicine at the Catholic University of Murcia. The activity took place in May 2024 and involved 52 participants (24 postgraduate nursing students and 28 postgraduate student physicians). A mixed-method descriptive study was conducted using a 16-item self-administered questionnaire. Data were analyzed using the Shapiro–Wilk test for normality, Fisher’s F test, and the Mann–Whitney U test to evaluate the relationship between variables (*p* < 0.05). Results: A total of 22 questionnaires were collected (16 from nurses postgraduate student and 6 from postgraduate physicians). Positive attitudes toward nursing stereotypes were found in 9 of the 13 items. No statistically significant differences were observed between the groups regarding most stereotypes, except for one. Negative stereotypes about nursing leadership, professional autonomy, and patient relations were more prominent among nursing students. Conclusions: Stereotypical perceptions exist among both postgraduate nursing and postgraduate student physicians, particularly in nursing leadership and autonomy. Most participants expressed satisfaction with the simulation-based IPE activity, indicating its value in improving the understanding of other professionals’ roles. IPE should be incorporated into health sciences education.

## 1. Introduction

Ensuring high-quality patient care necessitates collaboration among healthcare professionals. Interprofessional teams that exhibit mutual respect and effective communication enhance patient outcomes [1]. According to the World Health Organization (WHO) [2], interprofessional education (IPE) is an activity wherein health professionals engage with one another to achieve shared learning objectives concerning knowledge, skills, or attitudes, thereby fostering collaboration. The Centre for the Advancement of Interprofessional Education (CAIPE) similarly defines IPE as collaborative learning aimed at enhancing interprofessional cooperation.

Clinical practice is inherently interprofessional, with professionals working integratively to prioritize patient needs. Interprofessionalism encompasses the exchange of knowledge and practices across diverse perspectives, a process actively promoted through IPE at all educational levels [3,4,5]. The concepts of “Interprofessionalism” and “Teamwork” are closely interconnected, with effective teams significantly contributing to positive patient outcomes, as highlighted by Herrera-Miranda [6] and Schonhaut [7]. Irwing et al. (2012) [8] emphasize the importance of interdisciplinary collaboration, particularly in critical care settings, where research demonstrates favorable outcomes associated with interdisciplinary healthcare models.

Competence in teamwork minimizes errors and enhances patient safety by incorporating various healthcare professionals, notably physicians and nurses [9,10,11,12,13]. Research indicates that teamwork training improves team dynamics, clinical processes, and patient outcomes, particularly through models such as Crew Resource Management and TEAMSTEPPS, which enhance non-technical skills and safety [14,15,16,17,18,19,20,21,22].

Despite the emphasis on “soft skills” such as empathy, empirical evidence indicates that teamwork in healthcare remains inconsistent and often conflictual. While empathy plays a crucial role in breaking down barriers to effective teamwork, discrepancies persist between traditional practices and contemporary collaborative demands [23,24,25]. Limited research on interprofessional relationships between physicians and nurses reveals biases, methodological constraints, and a narrow focus on specific clinical contexts [26,27,28,29,30,31,32].

Hall [33] connect interprofessional empathy with teamwork skills, noting that collaborative behaviors vary across settings due to cultural and structural differences. Moreno-Leal et al. [34] identify frequent disruptive behaviors in high-stress healthcare environments, including belittling and lack of empathy, which contribute to the formation of interprofessional stereotypes. Quadflieg and Macrae (2011) [35] describe these stereotypes as cognitive frameworks encapsulating beliefs and expectations based on professional affiliations. Stereotypes shaped by hierarchical structures undermine empathy, cooperation, and teamwork, with physician dominance persisting despite advancements in nursing, thereby obstructing true interdisciplinarity [36,37,38,39,40,41,42,43,44,45,46].

The portrayal of social groups significantly shapes how their members interact with others and how they are perceived in return. In social psychology, stereotypes extend beyond mere traits, encompassing justifications for these characteristics and expectations of behavior, particularly in contexts with unequal group status or multiple group interactions [47,48].

In nursing, stereotypes can impede professional development and its recognition as a scientific discipline. A systematic review of current evidence highlights that nursing is often viewed as a feminine profession with limited competencies, low social status, modest salaries, and restricted autonomy in relation to the medical field [49]. Previous studies among non-nursing university students [50] and nursing students [51] reveal persistent stereotypes and self-perceptions that undermine the profession. Women in nursing are often seen as more caring than physicians but perceived as less independent and lacking leadership capabilities.

Empathy and professional attitudes play a pivotal role in mitigating negative interprofessional stereotypes. By fostering a genuine understanding of the experiences and contributions of colleagues from diverse disciplines, empathy reduces prejudices and promotes a collaborative mindset. This is particularly important in challenging notions of professional hierarchy, such as those often observed between physicians and nurses. Empathetic and respectful attitudes strengthen team cohesion by dispelling stereotypes about skills and roles, thereby enhancing interprofessional relationships.

Measuring stereotypes can provide insights into empathetic attitudes towards other professions. Furthermore, interprofessional training programs designed with this focus can significantly improve team dynamics by implementing measurable and practical strategies within healthcare environments.

Our aim was to examine the presence of ingrained stereotypical beliefs among nursing students and postgraduate student physicians engaged in interprofessional training activities focused on clinical simulation within the context of out-of-hospital emergency medicine. We also aim to analyze the perception of simulation as an interprofessional training strategy.

## 2. Materials and Methods

### 2.1. Study Design and Population

A pilot study was designed as a cross-sectional descriptive design investigation, involving participants from the official medical and nursing training programs at the Catholic University of Murcia (UCAM) during the 2023–2024 academic year. The findings from this pilot will serve as a foundation for future research, employing broader and more comprehensive study designs.

### 2.2. Participants and Sample

Study participants were recruited from the regulated medicine and nursing degree programs at UCAM. Among the offered courses, the Official University Master’s Degree in Urgent and Emergency Medicine and the Official Master’s Degree in Nursing in Emergencies and Special Care were selected for convenience. Participants were chosen based on the following criteria: (a) they must be students enrolled in the official emergency master’s programs offered by UCAM in the 2023–2024 academic year; (b) they must provide written informed consent on the first page of the questionnaire. This consent explicitly stated the need to maintain confidentiality regarding the activities conducted during the research period. No specific exclusion criteria were applied. Thus, a total of 52 individuals participated: 28 from the master’s program in medicine and 24 from the master’s program in nursing. This represents a selection of 93% of the enrolled students in these programs.

The Official University Master’s Degree represents a specialized training pathway formally recognized by healthcare and educational institutions, serving as the current de facto route for specialization among these professionals, pending formal specialization through residency programs. This training lasts for one academic year and is open to already-qualified professionals with clinical experience, who may remain actively employed while undertaking this training. Consequently, the content covered in these programs has direct applicability to daily clinical practice.

### 2.3. Study Design

A total of four interprofessional clinical simulation sessions were planned. Participants were divided into four subgroups, composed proportionally (the same number of subjects in each group) by clusters (medicine group vs. nursing group) with a random selection of the individuals within them. Members of each subgroup were randomly assigned to work teams as follows: two teams consisting of two postgraduate student physicians and one postgraduate nurse student (teams 1 and 2), one team composed of one postgraduate student physician and two postgraduate nursing students (team 3), and one team consisting of two postgraduate student physicians and two postgraduate nursing students (team 4). These groups and teams remained consistent throughout the training period.

Thus, the 52 participants were divided into four groups of 13 members each, comprising 7 postgraduate student physicians and 6 postgraduate nursing students. These subgroups were labeled A, B, C, and D, while the teams were generically designated as A1, A2, A3, and A4, consistent across all subgroups. This composition allows for the analysis of participant behaviors and attitudes based on the presence of peers and how this influences interprofessional relationships.

Four interprofessional training sessions were planned based on clinical simulations. Each session lasted 4 h and involved a clinical simulation instructor from the medicine program and another instructor from the nursing program. In each session, each of the four teams faced situations specifically designed for those sessions but presented randomly.

In sessions 1 and 2, scenarios and environments were developed for the management of non-clinical skills, which had been previously familiar to the participants. In sessions 3 and 4, participants dealt with clinical scenarios that were unfamiliar to them, focusing on themes relevant to the master’s program in Urgent and Emergency Care, applicable to both medicine and nursing.

Each group participated in these four sessions independently at different times and days. Thus, each team faced four distinct situations: two aimed at specifically developing soft skills and two addressing complex urgent care scenarios. All scenarios were based on the development of high-fidelity clinical simulation.

### 2.4. Structure of Clinical Simulations: Environment and Conditions

Each simulation session was designed with an effective duration of 4 h, dedicating one hour to each of the different scenarios planned for each team. A total of four sessions were developed, numbered from 1 to 4, with identical content for each subgroup.

In sessions 1 and 2, soft skills such as closed-loop communication, emotional support, situational leadership, and situational awareness were addressed, based on teamwork models like CRM (Crew Resource Management) and TEAMSTEPPS. Sessions 3 and 4 focused on complex clinical scenarios, including advanced CPR, management of a polytrauma patient (fall from height), management of a polytrauma patient (trapped in a vehicle), drowning management, management of intentional poisoning, management of a patient with acute coronary syndrome, management of an electrocution patient, and management of a patient trapped under heavy weight.

This resulted in 16 simulation sessions (4 for each subgroup), encompassing 4 distinct clinical simulation experiences (64 different simulation scenarios), each lasting one hour. The clinical simulations were conducted in the simulation facilities at UCAM’s Jerónimos campus, which are equipped with the necessary material and human resources (actors) to realistically simulate the planned critical scenarios. These facilities have hosted clinical simulations for over 15 years, as this methodology is integrated into the curricula of all health-related programs, particularly in the fields of nursing and medicine.

Each of the clinical simulation sessions were always led by two clinical simulation instructor-facilitators accredited by national organizations (Spanish Society of Clinical Simulation and Patient Safety—SESSEP), with experience in clinical training at the university as members of the Faculty of Nursing and the Faculty of Medicine of UCAM.

A nurse instructor and a physician instructor participated in each session. The instructors were not part of the study sample, as they were the researchers of this study (J.M.C.-P., A.R.-R., S.N.-C., and J.M.P.-R.).

The sessions were designed according to INACLS [52] standards, maintaining the traditional structure of briefing, scenario development, and subsequent debriefing, with a planned duration of 5-15-40 min, respectively. Given the content to be covered, the type of participants, and the resources to recreate actions (realistic elements such as recreated boxes, real ambulances, etc.), the sessions can be classified as Zone 3 simulations [53,54].

Entrance into the scenarios was conducted randomly at the beginning of each session. During the clinical scenarios performed by each selected team, the other members of the subgroups (the remaining teams) actively observed through a video-streaming system and a multiparameter patient monitor in differentiated adjoining rooms. If the scenario took place outdoors or within the ambulance school, the images were streamed to observers in the designated room.

Following each scenario, a debriefing was co-facilitated by both instructors, adhering to INACLS guidelines for debriefing [55], co-debriefing, and interprofessional education [56]. A debriefing model based on the PEARLS method was utilized [57,58,59].

### 2.5. Measurement Instruments

An ad hoc questionnaire was developed consisting of 19 items: 3 demographic questions (age, gender, and degree program) and 13 Likert-type items with 5 response options (ranging from 1 to 5, where 1 = Strongly Agree, 2 = Agree, 3 = Neutral, 4 = Disagree, and 5 = Strongly Disagree). These items assessed the most relevant stereotypes and preconceived notions in the physician–nurse relationship.

The questions related to the different stereotypes were further developed based on published research which discusses the presence of stereotypes and behavior, which show the presence of this type of ideas and behaviors. These questions were elaborated on jointly by A.R.R. and S.N.C. A consensus was reached with the rest of the researchers to establish the validity of the content. It should be noted that all the researchers have extensive clinical experience, so they have first-hand knowledge of the social reality of the relationship between physicians and nurses.

Additionally, three open-ended questions were included, asking participants about their feelings during the simulation sessions, the positive aspects they would highlight from the joint simulation experience, and the areas they believe could be improved, along with their reasons for these suggestions.

### 2.6. Data Collection

Upon the completion of the clinical simulation sessions, students were invited to participate via an online questionnaire. This questionnaire included informed consent clauses prior to participant responses. This questionnaire was accessed exclusively through a web link provided by the researchers to all students participating in the experience. The form only allowed the sending of a single response from the email address submitted. In this way, each subject who received an email inviting participation could send a single response. Participation was voluntary and did not involve any rewards.

From the time the link was sent, participants had 30 days to respond. Any responses submitted after this deadline were excluded from the data analysis. This time frame was established to minimize memory bias and capture impressions that were the closest to the experiences. Responses were collected between 18 April and 18 May 2024.

### 2.7. Data Analysis

Data were analyzed using SPSS v21.0 (IBM Corporation, Armonk, NY, USA). Participants’ perceptions regarding the thirteen items were analyzed using descriptive statistics (mean, frequency percentages, and standard deviations). To assess the relationships between the two studied populations (postgraduate nursing students and postgraduate student physicians), independent samples F-tests and Mann–Whitney U tests were conducted. The significance level was set at *p* < 0.05.

### 2.8. Ethical Considerations

Ethical approval for this study was obtained through the Ethics Committee of the Catholic University of Murcia (UCAM), protocol report number CE122201. Those who agreed to participate provided their informed consent to receive the link to the web questionnaire via email. Prior to completing the questionnaire, a text about the purpose of the questionnaire and the guarantee of privacy, the voluntary nature of their participation, and the willingness not to send the questionnaire without consequences were included. The acceptance of these conditions implied a signing of consent to participate in the study.

## 3. Results

### 3.1. Demographic and Likert Scale Data Analysis

After the latency period, responses were obtained from a total of 22 participants, resulting in an overall response rate of 42.3%. Among these, 16 were nursing students and 6 were postgraduate student physicians, reflecting response rates of 66.7% and 21.4% for each group, respectively.

Regarding gender, 68.2% (n = 15) were female and 31.8% (n = 7) were male. The age range of the sample was 22 to 47 years, with a mean age of 28 years. The detailed demographic data are presented in Table 1.

Appendix A present the descriptive statistical results of the responses provided by the studied participants for the thirteen items, analyzed for the total sample, as well as for the postgraduate nursing students and postgraduate student physicians groups, respectively.

If we analyze the data of all the participants without taking into account the different professions (Appendix A. All Participants) to which they belong, we find that means range from 1.45 (Item 12) to 4.27 (Item 13), with both positive and negative skewness, indicating the variability in responses by item. The mode values tend to align with the extreme high or low scores, suggesting polarized opinions. Kurtosis values suggest some data points with elongated distributions, particularly for items with extreme scores. A detailed analysis shows that eight of the items (items n° 3-4-5-7-8-9-10-13) show response averages higher than 3 or, in other words, disagreement with the statement made; on the contrary, the average is lower than 3 in the remaining five (items n° 1-2-6-11-12).

If we disaggregate the data, in the group of postgraduate nursing students (n = 16), (Appendix A (the postgraduate nursing student group)), the data trends observed in the general table are maintained, although the mean scores are sharper both in the scores higher than 3 (disagreement) and in the scores lower than 2 (agreement). The standard deviations are moderate in all items, reflecting homogeneity in responses, especially in empathy-related items.

The more clustered scores around values farther from 3 indicate that the homogeneity of the group’s responses is greater than those of the general group or those of the physicians; the lower variability (moderate standard deviations) indicates some consensus. Skewness is generally low, but Items 6 and 11 have a positive skew, suggesting nurses may respond with lower scores on those items. Negative kurtosis in several items indicates that the responses are spread, while Items 7 and 13 show a more peaked distribution, hinting at consistent positive views on these items.

On the contrary, in the group of postgraduate student physicians (n = 6), (Appendix A (the postgraduate student physician group)) eight items have means higher than 3 (disagreement with the item) and seven lower than 3 (agreement with the item). This indicates a difference in criteria with respect to the group of nursing students in two items in particular, item 3 and item 8. High kurtosis in specific items suggests dispersed responses and varied attitudes toward interprofessional collaboration. A few items show high kurtosis, suggesting concentrated responses, while negative kurtosis in others indicates a wider spread of responses. Positive skewness on Items 2, 11, and 12 points to a tendency for lower scores on these, while Items 3 and 9 show slight negative skewness, reflecting a higher agreement among participants on these statements.

The normality analysis using the Shapiro–Wilk test (Appendix A) indicates that the overall population does not follow a normal distribution statistically. The distribution of response frequencies given by participants for each item, disaggregated by group, is detailed in Appendix A.

When we analyze the data descriptively, based on the degree of agreement with the statements presented in the different items (Table 2), both groups predominantly respond in the same affirmative direction (either somewhat agreeing or disagreeing) for 9 out of the 13 items. In this way, participants consider that the work performed by nurses is physically demanding (77.3%) and mentally challenging (91%). They believe that “*nurses do not require less knowledge than doctors to perform their work*” (59.6% of participants); they consider that “*nursing shares the same field of knowledge as medicine*” (41%); they agree that “*caring and curing are not synonymous tasks*” (63.7%); they believe that “*showing sensitivity and kindness is part of nursing work*” (76.5%); they do not believe that “*nurses should show more empathetic attitudes than doctors*” (81.7%); they consider that “*nurses should not subordinate their work to medical decisions*” (40.9%); and 50% of participants do not consider that “*the natural leaders of the healthcare team are doctors*”. Regarding the statement that “*nursing decisions should be consulted with medicine*”, 40.9% of participants express uncertainty on this item. A large majority (90.9%) believe that “the nursing profession carries significant responsibility and is based on scientific evidence” (100%). Finally, more than 80% of respondents believe that “*nursing does not solely implement medical orders, which implies that it has a certain degree of autonomy*”.

The qualitative analysis (Table 3) of the direction of responses indicates that both groups agree on the sentiment expressed regarding preconceived notions about the professional relationship between physicians and nurses in 4 out of the 13 items; however, they differ in the remaining 9. It also shows that of the nine items reflecting the preconceived undervaluation of the nursing profession, six exhibit positive attitudes, while three maintain negative or erroneous attitudes in both groups.

A detailed descriptive analysis of these intergroup differences reveals that in item 4, “Nursing shares the same field of knowledge as Medicine”, 50% of postgraduate nursing students agree with this statement, while the same percentage of postgraduate student physicians express indecision, with 33.4% disagreeing—almost as many as the nurses who share this opinion.

Another point of disagreement can be found in item 8, “Nurses subordinates its work to the decisions of Physicians”. In this item, 66% of postgraduate nursing students disagree, while 50% of postgraduate student physicians agree; however, 25% of nurses express indecision, as do the other 50% of postgraduate student physicians. Notably, 18.7% of postgraduate nursing students align with 50% of postgraduate student physicians in affirming this item.

Regarding item 9, which states that “Natural leaders of the healthcare team are physicians”, 56.2% of postgraduate nursing students disagree with this statement, compared to 33.3% of postgraduate student physicians who agree. In this context, the physician group is completely divided, with one-third supporting it, one-third opposing it, and the remaining third expressing indecision.

Finally, in item 10, “Nursing decisions should be consulted with Medicine”, 43.7% of postgraduate nursing students express indecision regarding this statement, while 31% agree and only 24% disagree. Conversely, the physician group indicates a 50% disagreement with this statement and only a 16% agreement.

Finally, in light of the data provided by descriptive statistics and qualitative observation, we analyzed the relationship between the variables (Table 4) using Fisher’s test and Mann–Whitney U tests to examine the relationship between the professional group and the degree of agreement with each item. Despite both being considered qualitative cut-off variables, the Chi-square test was dismissed due to small marginal counts and expected data values of less than 5 in over 20% of the options for each item.

For the Fisher test, the items were grouped in three different ways to ensure a detailed analysis of all options. The first analysis excluded the “3” option (indecisive respondents), thus grouping responses 1 and 2 as “agree” and responses 4 and 5 as “disagree” with the proposed statement. The second analysis included responses 1, 2, and 3 in the “agree” group, while responses 4 and 5 were placed in the “disagree” category. The final analysis grouped responses 1 and 2 into the “agree” category and responses 3, 4, and 5 into the “disagree” group.

The analysis of the results indicates a wide majority absence of statistically significant relationships in the responses given between the groups, attributable to their belonging to the same profession; in all scenarios, the Fisher test yielded a significance greater than 0.05 (*p* > 0.05), thus rejecting the hypothesis of a relationship between the analyzed variables. Only in one of the items was there a statistically significant relationship: item 8, “Nursing subordinates its work to the decisions of Medicine”, where this relationship was slightly significant with *p*-values of 0.049, 0.046, and 0.019 for each of the analyzed scenarios.

Similarly, the Mann–Whitney values for the items compared between groups indicate no difference in responses attributable to the variable of the professional category to which the individual belongs. Like in the Fisher test, the only item with a significant difference between groups (*p* < 0.05) was item 8. The other items did not show statistically significant differences.

### 3.2. Analysis of Open-Ended Questions

Along with the thirteen previously discussed items, three open-ended questions were included, allowing participants to respond freely. These questions were as follows: (1) “*How did you feel during these joint simulations?*”; (2) “*What positive aspects would you highlight from this joint simulation experience?*”; and (3) “*What aspects do you think could be improved? Why*?”

Approximately 90% (n = 19) of the responses to the three questions contained positive evaluations of the activity, emphasizing the benefits of interprofessional education through high-fidelity clinical simulation scenarios for understanding each other’s professional roles and enhancing one’s own professional role, especially that of nursing in relation to medicine. Participants noted, for example, “*I realized the independence that nurses have compared to doctors*” (Subject 3), and “*It helps to understand each other’s working methods*” (Subject 11), as well as “*I learned what is expected of me and what I can expect from doctors in an emergency situation*” (Subject 13).

Participants highlighted the necessity of these experiences for developing teamwork competencies in a manner more reflective of actual healthcare settings: “*It is more realistic since it involves different professional profiles*” (Subject 11), and “*Training in an environment that more closely resembles what we will encounter in real life*” (Subject 2). They also mentioned, “*It helps to step out of the comfort zone we have with our colleagues and work with others, which will help us in our day-to-day work*” (Subject 4).

As a training method, interprofessional education through simulation allows participants to “*experience the distribution of roles more realistically*” (Subject 2). It enhances the psychosocial aspects of healthcare professionals such as empathy, teamwork, decision-making, and communication within the healthcare team (Subject 8). Participants noted, “*Everyone performs their usual role, and we can improve as a team*” (Subject 22). There were suggestions for role switching during some simulations to better understand the roles of other professionals, which could help to enhance professional empathy between groups: “*A simulation with role exchange could be implemented to gain another perspective*” (Subject 22).

Regarding aspects to improve, participants emphasized the need for these interprofessional simulations to be organized after independent training within each group. They pointed out the importance of having covered the theoretical content related to the scenarios beforehand: “*When the simulations were conducted, the medical colleagues hadn’t yet had classes on severe trauma, and I think it’s essential that we both have reviewed those concepts prior to the simulations*” (Subject 7).

Another suggested improvement was to focus scenario design on transversal competencies, such as soft skills and teamwork, rather than solely technical or skill-based competencies. “*I would propose simulated scenarios where both parties can participate at the same level, both medicine and nursing. In my case, most scenarios seemed more geared towards medicine than nursing. Addressing this would help facilitate good teamwork, where everyone is part of the whole and each participant plays an important role in the simulation*” (Subject 10).

## 4. Discussion

The first aspect to comment on, which marks the rest of the research results, is the low participation rate of the subjects. Of the 52 subjects who participated in the activity and were requested to complete the questionnaire, only 22 subjects submitted their responses after the deadline (30 days), resulting in a participation rate of just over 42% of the initially selected participants. As a corrective strategy, the response time was extended, and personal reminders were sent to the participants; despite this, no additional responses were received. This low participation rate, along with the disparity between the different population groups involved (16 postgraduate nursing students vs. 6 postgraduate student physicians), makes the data obtained in this study inconclusive and not significantly relevant. Nevertheless, similar studies have been conducted with a low level of participation [60] (N = 22; 14 nurses vs. 8 physicians).

The analysis of the research data yields several ideas that may be significant. First, an analysis of the “missing subjects” reveals a concerning fact: of the 30 subjects who did not participate by returning their questionnaires, 73.3% were postgraduate student physicians (n = 22) and 15.4% were postgraduate nursing students (n = 8). Similar studies show similarly low response rates [38,39], with global response rates around 40% (70–75% for nurses and 15–25% for physicians).

This difference can be attributed to the different sociological composition of the participating hospitals, where the nursing population is usually larger than the physicians population, or it may indicate a certain level of disinterest or apathy regarding participation, especially among the physician group.

A general overview of the published evidence in this field presents deficiencies regarding the composition of the studied subjects, with a significantly higher number of nurses participating compared to physicians, yet the sociological nature of this situation has not been analyzed. Generally speaking, such studies show an over-representation of nurses relative to physicians, with a ratio of between 2 and 3 nurses for every physician participating in the study. Studies like that of Elham and El-Hanafy [44] show a participation rate of 2.5 nurses for each physician in a study on perceived interprofessional relationships in a hospital; Tylor analyzed the perceptions between anesthetists and nurse anesthetists including 60 physicians and 230 nurses; Thomas et al. [40] included 90 physicians and 230 nurses; Elsous et al. [60] involved 176 nurses and 53 physicians; Amsalu et al. [61] included 308 nurses and 107 physicians.

It is also striking that in those items proposing statements that reaffirm or present biased or erroneous ideas ingrained in the collective subconscious regarding the nursing profession, the nursing group scores in favor of that negative thinking. For nurses to believe that “Nursing shares the same field of knowledge as Medicine”, that “It is part of the work of nursing to show sensitivity and kindness to patients’ feelings”, and that “Nursing decisions must be consulted with Medicine”, only serves to diminish the professional image of nursing and highlights not only the redundancy of a sociological problem but also an educational and formative one.

Educational approaches in nursing degrees must address these deviations in thoughts and attitudes that underlie the collective subconscious and contribute to empowering future nursing cohorts. Such educational activities can help to reduce these erroneous ideas and eliminate them from the mental framework of the nursing community. Combatting these attitudes cannot be achieved solely through classroom discourse; they must be practiced and made conscious through interprofessional activities, particularly those based on practical discussion, such as case study discussions or interprofessional clinical simulation.

These stereotypical ideas unfortunately continue to be established in nursing thought, leading, in some cases, to self-fulfilling prophecies [62].

Professions and professionals develop within a social context shaped by stereotypes that influence perceptions of the profession and affect its growth and development. Students, generally young individuals starting their nursing studies [63,64], largely begin with socially preconceived ideas based on stereotypes widely disseminated by the media through television series and movies [65].

Educational institutions aim to promote a discourse that fosters the change in these preconceived notions and stereotyped images. However, the clash with reality and the reinforcement of particularly negative stereotypes—such as limited autonomy, subordination of actions to medical decisions, professional sexism, and glass ceilings—influence both students in training [66] and already-graduated nurses [67,68], leading them to abandon the profession. This is a global issue, with the intention to leave the profession in the first year as a nurse estimated at 26% [69,70], although this cannot be attributed exclusively to these factors, as issues such as burnout, salary concerns, and psychological overload also play significant roles.

Analyzing the responses given by the nursing group regarding leadership, it is striking that only 50% of the respondents disagreed with the statement “The natural leaders of the healthcare team are doctors”, while 25% agreed with it. One would expect the opposite, as their training program has encouraged this aspect. In this context, one-third of the student physician group supports this statement while another third opposes it (indicating a shared leadership), which is surprising given the expectation of a higher percentage favoring the physician as the central leader of the healthcare team. This once again points to the presence of negative attitudes among the nursing group and a positive attitude from the student physician group regarding nursing competencies.

The idea that healthcare team leadership falls upon physicians is a traditionally held conception in society and has led to many health professionals to perceive doctors as leaders in patient care for a long time, while nurses play a subordinate role; this view has been shared by nurses themselves [71]. This role appears to be changing at the administrative and management levels specifically. However, in interprofessional relationships and teamwork, clinical nurse leadership is viewed as a marginal or exceptional situation, although this is fortunately evolving [72,73].

This context makes assuming the role of clinical leadership seem less natural for nurses. In the traditional hierarchy, nurses typically occupy a more subordinate role, which discourages them from questioning or deviating from rules and regulations or seeking a leadership role, even when the purpose is to benefit the patient. As a result, they play a marginal role in clinical leadership and feel more comfortable ceding that leadership to physicians [74].

These negative attitudes or stereotypes regarding the nursing profession detected among the interviewed doctors, such as “It is part of nursing to show sensitivity and kindness to patients’ feelings”, “Nursing subordinates its work to medical decisions”, and “The natural leaders of the healthcare team are doctors”, align with expectations given the social climate and stereotypes that, as elements of society, are not unfamiliar to them. However, in general, the student physician group displays more positive attitudes than expected, in contrast to the findings within the nursing group. In this regard, diverse evidence indicates positive preconceptions from the student physician group towards the nursing student group [75,76].

All these stereotypes can be countered through interprofessional education by implementing integrative activities that not only detect shortcomings in the mental framework of different professionals but also facilitate the discussion and shared correction of these erroneous preconceptions [77].

The idea is that when health professionals understand each other’s roles, it stimulates effective communication and teamwork, thereby improving the quality of care provided to patients. The World Health Organization (WHO) recommends integrating interprofessional education as an essential part of the curriculum for health-related professions at the undergraduate level.

An early intervention contributes to social demystification, the correction of erroneous ideas and behaviors observed in clinical environments during undergraduate training, and the formal structuring of mental frameworks that can withstand the social pressure upon entering the workforce [78]. This is the only way to foster the necessary social transformation.

The study participants’ perceptions of their involvement in the experience reveal a high level of satisfaction and its usefulness for developing teamwork skills. Evidence supports this notion, confirming high satisfaction among students who participate in such activities. Similarly, clinical simulation is a particularly useful methodology for implementing interprofessional education strategies, making its implementation not only feasible but desirable [79,80,81,82].

The study participants’ perceptions of their involvement in the experience reveal a high level of satisfaction and its usefulness in developing teamwork skills. This idea is corroborated by the evidence, which shows a high level of satisfaction among students who participate in this type of activity. Likewise, clinical simulation is a particularly useful methodology for implementing interprofessional education strategies, so its implementation is not only feasible, but also desirable [82,83]. In this sense, the available evidence indicates the need to incorporate this type of activities in the curricula of health sciences degrees, especially in nursing studies, improving substantial aspects for interprofessional relations such as communication and teamwork [84,85,86,87,88,89].

## 5. Implications

The presence of stereotypes and negative professional attitudes remains prevalent among nursing and medical students. Therefore, it is essential to include shared workspaces where both groups can interact and engage in inclusive experiences, enabling them to recognize the true roles and value of each profession. By fostering mutual understanding between these groups, it becomes possible to challenge and modify these misguided perceptions. The undergraduate and graduate stages of nursing and medical education are particularly crucial for this type of training, as the malleability of ideas during these formative years makes these experiences especially impactful.

Promoting interprofessional education experiences based on clinical simulation—structured, planned, and integrated into the curricula of both degree programs—not only enhances well-known competencies such as teamwork and collaboration but also addresses the stereotypes associated with these professions.

Therefore, we suggest further research in this line of intervention, given the potential present and future benefits for the social collective of the health professions.

## 6. Limitations

We acknowledge the use of non-validated questions, a small number of subjects studied, and the heterogeneous composition of the population groups in this study, as well as a low response rate and a need for improved data collection and follow-up strategies. Nevertheless, the research objectives have been met.

## 7. Conclusions

Stereotypical perceptions regarding the role and autonomy of nursing, as well as leadership in the health sector, persist among postgraduate nursing and student physicians. Despite educational efforts to empower nursing professionals, preconceived ideas limiting their professional development and affecting interprofessional dynamics have been identified. It is concerning that a significant proportion of participants adhere to statements that perpetuate the subordinate image of nursing, such as the belief that “natural leaders of the healthcare team are physicians” and that “nurses decisions must be consulted with physicians”. These attitudes reflect the persistence of social stereotypes and the urgent need to review and reform educational approaches in the health field.

However, it is encouraging that most participants expressed high satisfaction with the interprofessional simulation experience (IPE). This level of satisfaction suggests that simulation-based activities are effective tools for improving understanding of the roles and responsibilities of other health professionals. Participants’ comments emphasized the value of these experiences in fostering more collaborative teamwork, highlighting the importance of understanding each other’s roles in real clinical situations. This positive appreciation of IPE indicates that students recognize the relevance of these activities in their training and professional development.

Therefore, it is crucial to systematically integrate interprofessional education into health sciences curricula. This will not only help to mitigate negative stereotypes but also strengthen collaboration between physicians and nurses, promoting a more holistic approach to patient care. Implementing IPE in health education will enable future professionals to develop essential communication, empathy, and teamwork skills necessary for providing patient-centered care.

In conclusion, the integration of interprofessional education, particularly through methodologies such as clinical simulation, is fundamental to transforming stereotypical perceptions and improving the working dynamics among various health professionals. This transformation will benefit not only the professionals themselves but also positively impact the quality of care provided to patients, contributing to a more efficient and equitable healthcare system.

## Figures and Tables

**Table 1 healthcare-12-02449-t001:** Demographic distribution.

	Gender		Age					
	Women	Men	Mean	Median	Mode	ST	Max	Min
Postgraduate nursing students	11 (68.7%)	5 (31.2%)	26	25	23	4.73	42	22
Postgraduate student physicians	4 (66.6%)	2 (33.3%)	33.3	31	31	6.97	47	27
Total	15 (68.18%)	7 (31.8%)	28	26	23	6.22	47	22

**Table 2 healthcare-12-02449-t002:** Distribution of responses by level of agreement and group.

	Postgraduate Nursing Students n = 16		Postgraduate Student Physicians n = 6	
1. The work performed by nursing is physically demanding.	agree entirely	4	agree entirely	0
	agree	8	agree	5
	neither agree nor disagree	4	neither agree nor disagree	1
	disagree	0	disagree	0
	disagree entirely	0	disagree entirely	0
2. The work performed by nursing is mentally demanding.	agree entirely	9	agree entirely	1
	agree	6	agree	4
	neither agree nor disagree	1	neither agree nor disagree	0
	disagree	0	disagree	1
	disagree entirely	0	disagree entirely	0
3. The knowledge required by nurses to perform its work is inferior to that required by Physisians.	agree entirely	0	agree entirely	2
	agree	4	agree	0
	neither agree nor disagree	2	neither agree nor disagree	1
	disagree	8	disagree	2
	disagree entirely	2	disagree entirely	1
4. Nurses shares the same field of knowledge as Physisians.	agree entirely	1	agree entirely	0
	agree	7	agree	1
	neither agree nor disagree	2	neither agree nor disagree	3
	disagree	6	disagree	1
	disagree entirely	0	disagree entirely	1
5. Caring and curing are synonymous tasks.	agree entirely	2	agree entirely	0
	agree	4	agree	0
	neither agree nor disagree	1	neither agree nor disagree	1
	disagree	5	disagree	1
	disagree entirely	4	disagree entirely	4
6. It is part of nursing’s work to show sensitivity and kindness towards patients’ feelings.	agree entirely	7	agree entirely	2
	agree	6	agree	2
	neither agree nor disagree	2	neither agree nor disagree	2
	disagree	1	disagree	0
	disagree entirely	0	disagree entirely	0
7. Nurses must exhibit more empathetic attitudes than Physisians.	agree entirely	1	agree entirely	1
	agree	1	agree	0
	neither agree nor disagree	1	neither agree nor disagree	0
	disagree	8	disagree	4
	disagree entirely	5	disagree entirely	1
8. Nurses subordinates its work to the decisions of Physisians.	agree entirely	0	agree entirely	0
	agree	3	agree	3
	neither agree nor disagree	4	neither agree nor disagree	3
	disagree	8	disagree	0
	disagree entirely	1	disagree entirely	0
9. The natural leaders of the healthcare team are the Physisians.	agree entirely	1	agree entirely	0
	agree	3	agree	2
	neither agree nor disagree	3	neither agree nor disagree	2
	disagree	3	disagree	2
	disagree entirely	6	disagree entirely	0
10. Nurses decisions must be consulted with Physisians.	agree entirely	0	agree entirely	0
	agree	5	agree	1
	neither agree nor disagree	7	neither agree nor disagree	2
	disagree	2	disagree	3
	disagree entirely	2	disagree entirely	0
11. The nursing profession holds significant responsibility.	agree entirely	9	agree entirely	3
	agree	6	agree	2
	neither agree nor disagree	1	neither agree nor disagree	0
	disagree	0	disagree	1
	disagree entirely	0	disagree entirely	0
12. The work of nursing is grounded in scientific evidence.	agree entirely	10	agree entirely	2
	agree	6	agree	4
	neither agree nor disagree	0	neither agree nor disagree	0
	disagree	0	disagree	0
	disagree entirely	0	disagree entirely	0
13. Nurses applies only the orders from Physisians.	agree entirely	0	agree entirely	1
	agree	1	agree	0
	neither agree nor disagree	1	neither agree nor disagree	1
	disagree	7	disagree	2
	disagree entirely	7	disagree entirely	2

**Table 3 healthcare-12-02449-t003:** Direction of response * and evaluative value of items by group.

	Postgraduate Nursing Students					Postgraduate Student Physicians				
	Negative Value		3	Positive Value		Negative Value		3	Positive Value	
Item 1				**→**					**→**	
Item 2				**→**					**→**	
Item 3				**→**					**→**	
Item 4	**←**								**→**	
Item 5				**→**					**→**	
Item 6	**←**					**←**				
Item 7				**→**					**→**	
Item 8				**→**		**←**				
Item 9				**→**		**←**				
Item 10	**←**								**→**	
Item 11				**→**					**→**	
Item 12				**→**					**→**	
Item 13				**→**					**→**	

* Sense of response. Arrow to the left: negative concept; arrow to the right: positive concept.

**Table 4 healthcare-12-02449-t004:** Statistical tests of relationships between variables. U Mann–Whitney and Fisher tests.

Item.	U Mann–Whitney	Fisher ^(1-2//3-4-5)^	Fisher ^(1-2-3//4-5)^	Fisher ^(1-2//4-5)^
1. The work performed by nursing is physically demanding.	--	--	--	--
2. The work performed by nursing is mentally demanding.	27.5 (0.102)	0.287	0.065	0.071
3. The knowledge required by nurses to perform its work is inferior to that required by Physisians.	56 (0.559)	0.334	0.348	0.305
4. Nurses shares the same field of knowledge as Physisians.	35.5 (0.354)	0.333	0.655	0.576
5. Caring and curing are synonymous tasks.	24 (0.072)	0.133	0.351	0.26
6. It is part of nursing’s work to show sensitivity and kindness towards patients’ feelings.	41 (0.609)	1	1	1
7. Nurses must exhibit more empathetic attitudes than Physisians.	54 (0.654)	1	1	1
8. Nurses subordinates its work to the decisions of Physisians.	**76.5** **(0.03)**	**0.049**	**0.046**	**0.019**
9. The natural leaders of the healthcare team are the Physisians.	63 (0.272)	0.334	0.635	0.581
10. Nurses decisions must be consulted with Physisians.	37.5 (0.437)	0.266	0.137	0.105
11. The nursing profession holds significant responsibility.	42.5 (0.679)	0.169	0.065	0.071
12. The work of nursing is grounded in scientific evidence.	--	--	--	--
13. Nurses applies only the orders from Physisians.	58.5 (0.428)	0.273	0.169	0.25

## Data Availability

The original contributions presented in the study are included in the article/Appendix A, further inquiries can be directed to the corresponding author.

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
