# Peer review of "Measuring Stereotypes in Interprofessional Education: A Pilot High-Fidelity Simulation Study Among Postgraduate Nursing and Physician Students in a Spanish University"

_healthcare, 2024, doi:10.3390/healthcare12232449_

Round 1

Reviewer 1 Report

Comments and Suggestions for Authors

The manuscript entitled ‘Impact of Stereotypes in Postgraduate Nursing and Medical Students: The Role of High-Fidelity Simulation in Interprofessional Education’ detailed an interprofessional training in a high fidelity simulation environment between postgraduate nursing and medical students. A survey was administered post-IPE to evaluate the satisfaction and potential stereotyping amongst the students. 

The title does not clearly reflect the actual study. The number of students who have completed the survey was very small with only 6 medical students completed the survey. It is difficult to assess the impact of stereotypes in two different groups of health professional students with such a small sample size. The small sample size is a significant limitation, as it precludes meaningful conclusions to be drawn for this research question.

The introduction covers the definition and concept of IPE as well as teamwork extensively. The introduction can be condensed and focus on the key point i.e. stereotyping of health professionals to provide a clearer and more concise narrative.

It was stated that the objective of this study was to “identify the extent of ingrained stereotypical ideas and the impact of Interprofessional Training on the levels of knowledge, attitudes, and empathy towards other professionals among postgraduate nursing and medical students” . The levels of knowledge and empathy were not objectively assessed in this study hence this shouldn’t be included in the objective. In fact, there are two different objectives - 1. To identify the extent of ingrained stereotypical ideas in nursing and medical students; and 2. To understand the attitudes of nursing and medical students towards a high fidelity simulation in IPE.

For objective 2, the authors need to expand on how their high fidelity simulation in IPE study is situated in the body of literature – why is there a need for this work when there are many other articles on high fidelity simulation in IPE between nursing and medical students published previously?

Duplicate references – 2 & 3, 73 & 74

The study design needs to reconsidered as the findings from this study are based on a single ad hoc survey which has not undergone validity and reliability testings, and the conclusions were drawn using data from a small sample size. In-depth interviews may be better suited to meet the primary objective of this study, which is to assess the impact of stereotypes in postgraduate Nursing and Medical Students after completing interprofessional education.

The results section requires significant reworking. Tables 2-6 are not required and should be included in supplementary date. Presenting the normality analysis for each item in the main result section detract the readers from focusing on the main findings. Similarly, presenting the mean, median, mode etc for all items for a study with such a small sample size does not provide much value, and disrupts the flow of the presentation.

There is a potential for the survey to be used as an instrument to assess the impact of stereotypes in different groups of health professional students, however this will require a significant amount of further work to establish the validity (face, content, construct) and reliability (test-retest and internal consistency) of the instrument.

Author Response

The manuscript entitled ‘Impact of Stereotypes in Postgraduate Nursing and Medical Students: The Role of High-Fidelity Simulation in Interprofessional Education’ detailed an interprofessional training in a high fidelity simulation environment between postgraduate nursing and medical students. A survey was administered post-IPE to evaluate the satisfaction and potential stereotyping amongst the students. 

The title does not clearly reflect the actual study. The number of students who have completed the survey was very small with only 6 medical students completed the survey. It is difficult to assess the impact of stereotypes in two different groups of health professional students with such a small sample size. The small sample size is a significant limitation, as it precludes meaningful conclusions to be drawn for this research question.

RESPONSE 

Thank you for your comments. The title has been modified accordingly. Regarding your comment on the sample size, we already recognize this as a limitation (lines 500-503), particularly the low number of medical students who participated in the study. It is likewise commented on in lines 394-403. We recognize that the total sample of 23 participants, and in particular the 6 physicians, could represent a limitation in generalizing the findings. Nevertheless, we would like to present some considerations regarding our methodology and the value of the results obtained, even with this reduced sample size.

-Observational design: This study presents an observational analysis, with an eminently exploratory character, focused on identifying the potential of interprofessional education (IE) through clinical simulation to address stereotypes among health professionals, an area where research is still emerging. Although we recognize that a larger sample could provide more conclusive data, these initial results offer a basis for identifying patterns that can guide future studies with greater representativeness.

- Similar precedents. Similar studies, show that multiprofessional involvement is extremely difficult. Scotten, Mitzi et al (2015) analyze the impact of interprofessional training and attitudes of healthcare professionals, in relation to communication, and measured on the satisfactions of patients who were jointly cared for. This may indicate that the interest of the different professional groups in participating in non-technical skills development activities is low, and therefore the effort of researchers and teachers should be high in order to achieve high participation rates.

In our case, the measurement activity was carried out at the end of the 4-week academic activity, in May, which was scheduled last in terms of teaching planning for the master's degree in Medicine and Emergency Nursing. That is to say, the participating students began their master's academic training in September 2023 and finished this course in May 2024; as it was the last course of the course and the collection of information was voluntary, we understand that the response rate was low, due to the low motivation and reward, as well as the desire of the participating students to finish their studies.

Regarding the low proportion of participants from the medical community, similar studies show data on medical participation similar to ours, where the proportion of nurses is higher than that of physicians. He, Song et al. 2024.

The introduction covers the definition and concept of IPE as well as teamwork extensively. The introduction can be condensed and focus on the key point i.e. stereotyping of health professionals to provide a clearer and more concise narrative.

RESPONSE

Thank you for your appreciation. Following this recommendation, we synthesized the text of the introduction, condensing information and thematic groups of evidence. We have reduced the length of the introduction by approximately 30%.

It was stated that the objective of this study was to “identify the extent of ingrained stereotypical ideas and the impact of Interprofessional Training on the levels of knowledge, attitudes, and empathy towards other professionals among postgraduate nursing and medical students” . The levels of knowledge and empathy were not objectively assessed in this study hence this shouldn’t be included in the objective. In fact, there are two different objectives - 1. To identify the extent of ingrained stereotypical ideas in nursing and medical students; and 2. To understand the attitudes of nursing and medical students towards a high fidelity simulation in IPE.

For objective 2, the authors need to expand on how their high fidelity simulation in IPE study is situated in the body of literature – why is there a need for this work when there are many other articles on high fidelity simulation in IPE between nursing and medical students published previously?

RESPONSE

Thank you for your comment. Certainly, the interprofessional empathy and the attitudes towards the other group are not measured. We eliminated them as an objective, and focused on steotypes as the main object of study.
However, we would like to emphasize that professional empathy and attitudes are key to reducing negative interprofessional stereotypes. Empathy enables professionals to understand and value the experiences and contributions of colleagues from different disciplines, reducing preconceived judgments. This favors an open and collaborative attitude, essential to counteract stereotypes such as the perception of professional supremacy (as between doctors and nurses). Empathetic and respectful attitudes create a climate of cooperation, where stereotypical ideas about skills and roles are reduced, promoting positive and functional interprofessional relationships in the health team.
Therefore, we consider that the measurement of steotypes may reflect empathic attitudes towards the other profession. And interprofessional training programs could contribute to all these easily measurable constructs.

Duplicate references – 2 & 3, 73 & 74

RESPONSE

Indeed, these are duplicities; the typo is corrected. Thank you for your appreciation.

The study design needs to reconsidered as the findings from this study are based on a single ad hoc survey which has not undergone validity and reliability testings, and the conclusions were drawn using data from a small sample size. In-depth interviews may be better suited to meet the primary objective of this study, which is to assess the impact of stereotypes in postgraduate Nursing and Medical Students after completing interprofessional education.

RESPONSE

As previously mentioned, this is an observational and exploratory study. The qualitative design is taken into account as a next phase of this work, to be developed in the current academic year 204-25, with another group of students. Certainly, the qualitative approach could provide an additional focus that would help to shed light on different aspects of this field. This work serves as a starting point to initiate discussion topics.

The results section requires significant reworking. Tables 2-6 are not required and should be included in supplementary date. Presenting the normality analysis for each item in the main result section detract the readers from focusing on the main findings. Similarly, presenting the mean, median, mode etc for all items for a study with such a small sample size does not provide much value, and disrupts the flow of the presentation.

RESPONSE

Thank you for your comments. The results have been rewritten and the tables have been moved to the annexes section.

There is a potential for the survey to be used as an instrument to assess the impact of stereotypes in different groups of health professional students, however this will require a significant amount of further work to establish the validity 

RESPONSE

Thank you for your consideration. As I have commented in previous answers, this is a first exploratory phase of further work; where the realization of focus groups is contemplated, from which, to elaborate questionnaires endowed with sufficient metric properties, with construct validity, content, measurement on a pilot group, with test-retest and finally a large multiprofessional sample. All this is part of a research project carried out within the group of researchers of the university, and in a doctoral program.

Reviewer 2 Report

Comments and Suggestions for Authors
  • Does the introduction provide sufficient background and include all relevant references?

    • Can be improved: The introduction covers important background information regarding stereotypes in healthcare and Interprofessional Education (IPE), but it could be expanded to include more recent references and a broader discussion of similar studies in other contexts.
  • Is the research design appropriate?

    • Yes: The mixed-method design is suitable for exploring both quantitative and qualitative aspects of the study. However, more consideration could be given to addressing the low response rate and ensuring balanced participation across both groups.
  • Are the methods adequately described?

    • Yes: The methods are generally well-described, including the simulation setup, participant groupings, and the questionnaire used. The study design and data collection are sufficiently detailed.
  • Are the results clearly presented?

    • Can be improved: The results are presented clearly, but more in-depth discussion of the statistical significance (or lack thereof) would be helpful. Tables are used to display data, but some interpretations could be expanded to better highlight key findings.
  • Are the conclusions supported by the results?

    • Yes: The conclusions align with the results, particularly in noting the persistence of stereotypes and the positive impact of IPE through simulation. However, given the low sample size and response rate, some of the broader conclusions should be approached with caution.
Comments on the Quality of English Language

The English is understandable, but there are some awkward phrasing and minor grammar issues that could benefit from light editing for clarity and readability.

Author Response

Does the introduction provide sufficient background and include all relevant references?

Can be improved: The introduction covers important background information regarding stereotypes in healthcare and Interprofessional Education (IPE), but it could be expanded to include more recent references and a broader discussion of similar studies in other contexts.

RESPONSE

Thank you for your feedback. According to the recommendations given by another reviewer, the introduction has been synthesized in its length, bringing together the information given and the blocks of evidence that support it.
However, following his recommendation, two references that were obsolete, 2005 (ref nº5) and 2000 (ref nº27), have been replaced by more updated references without altering the content of the introduction.

Is the research design appropriate?

Yes: The mixed-method design is suitable for exploring both quantitative and qualitative aspects of the study. However, more consideration could be given to addressing the low response rate and ensuring balanced participation across both groups.

RESPONSE

Thank you for your comments.
Regarding the low participation rate, we agree with your opinions, and we indicated it at different times in the discussion, as a limitation of our work. In this regard, I would like to add that similar studies show that multiprofessional involvement is extremely difficult. Scotten, Mitzi et al (2015) analyze the impact of interprofessional training and attitudes of health professionals, in relation to communication, and measured on the satisfaction of patients who were jointly cared for. This may indicate that the interest of the different professional groups in participating in non-technical skills development activities is low, and therefore the effort of researchers and teachers should be high in order to achieve high participation rates.
In our case, the measurement activity was carried out at the end of the 4-week academic activity, in May, which was scheduled last in terms of teaching planning for the master's degree in Medicine and Emergency Nursing. That is to say, the participating students began their master's academic training in September 2023 and finished this course in May 2024; since it was the last course of the course, and the collection of information was voluntary, we understand that the response rate was low, due to the low motivation and reward, as well as the desire of the participating students to finish their studies.

Are the methods adequately described?

Yes: The methods are generally well-described, including the simulation setup, participant groupings, and the questionnaire used. The study design and data collection are sufficiently detailed.

RESPONSE

Thank you for your appreciation. You are very kind.
This work has been conceived as the first of a series of works to be developed in the inteprofessional work group of the university, in different academic years. Hence its observational and exploratory nature. Future works will approach the problem from a qualitative point of view by means of focus groups, based on the topics that can be extracted from this work; at the same time, in another phase, the elaboration of a validated questionnaire made up of items from this work and the future conclusions of the focus groups will be approached; it will be piloted in a small group and demonstrated in a larger group. But these will be future phases of research.

Are the Results clearly presented?

Can be improved: The results are presented clearly, but more in-depth discussion of the statistical significance (or lack thereof) would be helpful. Tables are used to display data, but some interpretations could be expanded to better highlight key findings.

RESPONSE

Thank you for your feedback. Following your recommendations and those of other reviewers, the dense data tables have been removed from the original manuscript and moved to supplementary data, and a summary of the most salient findings and their interpretations has been added in their place. This brings clarity to the reader, and helps guide the reader to a better interpretation of the findings.

Are the conclusions supported by the results?

Yes: The conclusions align with the results, particularly in noting the persistence of stereotypes and the positive impact of IPE through simulation. However, given the low sample size and response rate, some of the broader conclusions should be approached with caution.

RESPONSE

Thank you for your comments. We are aware of the limitations of the study, as it has a limited sample size and a low representativeness of the medical community. We address these limitations in the discussion section (lines 394-403 and lines 500-503).

Reviewer 3 Report

Comments and Suggestions for Authors

Dear authors,

Thank you for the opportunity to review this manuscript, which highlights the role of teamwork in reducing human error and improving patient outcomes through interprofessional collaboration. I have some corrections:

INTRODUCTION: To start the introduction in a more formal tone, consider using: "Ensuring effective and high-quality patient care in the modern healthcare system requires close collaboration among the various healthcare professionals involved." The English should be refined for clarity and flow. For example, instead of "In this regard," a more scientific phrasing would be: "Regarding this issue, some authors consider that effective teamwork is a vital component for minimizing human error." Additionally, Phrases such as "It is here that..." can be adjusted to maintain a more formal and precise tone.

 RESULTS: In this section, as well as in the abstract, the tables refer to Nurses and Physicians, but it is actually about nursing students and medical students. Please correct this to reflect the correct terminology. Table 1 contains redundant information. Retain only the Mean and SD values for clarity. Tables 2-5 are not necessary and can be removed.

For Table 6, ensure that the items are described clearly. Please correct the use of 1++ 2+ as this notation is unclear.

In Table 7, only keep the Mean (SD)

Comments on the Quality of English Language

English language should be corrected by native speaker.

Author Response

Thank you for the opportunity to review this manuscript, which highlights the role of teamwork in reducing human error and improving patient outcomes through interprofessional collaboration. I have some corrections:

 INTRODUCTION: To start the introduction in a more formal tone, consider using: "Ensuring effective and high-quality patient care in the modern healthcare system requires close collaboration among the various healthcare professionals involved." The English should be refined for clarity and flow. For example, instead of "In this regard," a more scientific phrasing would be: "Regarding this issue, some authors consider that effective teamwork is a vital component for minimizing human error." Additionally, Phrases such as "It is here that..." can be adjusted to maintain a more formal and precise tone.

RESPONSE

Thank you for your feedback.
According to the recommendations of other reviewers, the introduction has been synthesized, and the level of English and level of writing have been revised to a more formal level. We have eliminated or modified colloquial expressions or expressions that invite this type of interpretation.

RESULTS: In this section, as well as in the abstract, the tables refer to Nurses and Physicians, but it is actually about nursing students and medical students. Please correct this to reflect the correct terminology. Table 1 contains redundant information. Retain only the Mean and SD values for clarity. Tables 2-5 are not necessary and can be removed.

For Table 6, ensure that the items are described clearly. Please correct the use of 1++ 2+ as this notation is unclear.

In Table 7, only keep the Mean (SD)

RESPONSE

Thank you for your comments. Accordingly, the terms “nurses” and “physicians” have been replaced by “postgraduate nursing students” and “postgraduate medical students”. However, I must mention that since the postgraduate level is reached after completing the undergraduate training of each of the degrees, the postgraduate students are all professionals in active practice of their corresponding profession; in fact, all the students are already in the labor market, developing their professional activity without any restriction whatsoever. Hence, they are considered as professionals and not as students.

Thank you for your feedback. The tables have been modified according to your recommendations. However, they have been removed from the manuscript, while a description of their main results has been introduced. Tables 1 to 6 will be attached as supplementary data, in a parallel file.

Reviewer 4 Report

Comments and Suggestions for Authors

The article fits within the thematic scope of the journal; however, it contains some significant shortcomings. Below, I present a few comments:

The first comment concerns the article's title. As suggested by the content of the article, the research pertains to only one academic institution. From the perspective of scientific integrity and methodological requirements, the title should at least specify the country (Spain) where the research was conducted. In its current form, the title might misleadingly suggest that the study has an international scope.

In the Introduction, the authors devote considerable space to ‘Interprofessional Education’ but do not define or explain how they understand one of the key concepts of the article - "stereotypes." The term "stereotypes" is ambiguous in the literature. It is not at all clear how the authors interpret the word "stereotypes." Moreover, it would be necessary to at least generally specify what type of stereotypes the authors intend to 'analyze' or describe in the article.

Main weaknesses of the article: First, the article concerns research conducted on a small sample of respondents (52 participants). Second, the research is limited to only one research institution. While it indicates certain trends within this particular institution, it is doubtful that the findings can be considered representative, as they are only regional in nature.

The article should include a section titled "Limitations," stating that the presented research is regional because it was conducted in only one academic institution.

Author Response

The article fits within the thematic scope of the journal; however, it contains some significant shortcomings. Below, I present a few comments:

The first comment concerns the article's title. As suggested by the content of the article, the research pertains to only one academic institution. From the perspective of scientific integrity and methodological requirements, the title should at least specify the country (Spain) where the research was conducted. In its current form, the title might misleadingly suggest that the study has an international scope.

RESPONSE

Dear reviewer. Thank you for your comments. We have added the country where the study was conducted in the title.

In the Introduction, the authors devote considerable space to ‘Interprofessional Education’ but do not define or explain how they understand one of the key concepts of the article - "stereotypes." The term "stereotypes" is ambiguous in the literature. It is not at all clear how the authors interpret the word "stereotypes." Moreover, it would be necessary to at least generally specify what type of stereotypes the authors intend to 'analyze' or describe in the article.

RESPONSE

Thank you for your appreciation. We had taken the concept for granted because of its ordinariness. A paragraph and two significant references have been added.

Main weaknesses of the article: First, the article concerns research conducted on a small sample of respondents (52 participants). Second, the research is limited to only one research institution. While it indicates certain trends within this particular institution, it is doubtful that the findings can be considered representative, as they are only regional in nature.

RESPONSE

Dear reviewer. We agree with your appreciations. We must remember that the study has a descriptive-exploratory character, and is intended to be the first step in a series of future works to be developed during the current academic year 2024-25 and following years, dealing with this topic from a qualitative perspective based on focus groups, to subsequently design a grid of questions to be validated according to the standards of scale creation: metric properties, internal and construct validation, pilot testing and finally large group testing. All this will be part of the doctoral program.

The article should include a section titled "Limitations," stating that the presented research is regional because it was conducted in only one academic institution.

RESPONSE

Although we do not address this section explicitly, because it is not journal policy, in the discussion section (lines 394-403 and lines 500-503), we recognize and address these limitations.

Reviewer 5 Report

Comments and Suggestions for Authors

This paper is a valuable contribution to the differences in perception of nurses and Drs. The introduction is very comprehensive. 

Author Response

This paper is a valuable contribution to the differences in perception of nurses and Drs. The introduction is very comprehensive. 

RESPONSE

Thank you very much for your words
I attach the latest version with the modifications that the reviewers have kindly made to the first manuscript submitted.

Round 2

Reviewer 1 Report

Comments and Suggestions for Authors

Author's Notes

The manuscript entitled ‘Impact of Stereotypes in Postgraduate Nursing and Medical Students: The Role of High-Fidelity Simulation in Interprofessional Education’ detailed an interprofessional training in a high fidelity simulation environment between postgraduate nursing and medical students. A survey was administered post-IPE to evaluate the satisfaction and potential stereotyping amongst the students.

The title does not clearly reflect the actual study. The number of students who have completed the survey was very small with only 6 medical students completed the survey. It is difficult to assess the impact of stereotypes in two different groups of health professional students with such a small sample size. The small sample size is a significant limitation, as it precludes meaningful conclusions to be drawn for this research question.

RESPONSE

Thank you for your comments. The title has been modified accordingly. Regarding your comment on the sample size, we already recognize this as a limitation (lines 500-503), particularly the low number of medical students who participated in the study. It is likewise commented on in lines 394-403. We recognize that the total sample of 23 participants, and in particular the 6 physicians, could represent a limitation in generalizing the findings. Nevertheless, we would like to present some considerations regarding our methodology and the value of the results obtained, even with this reduced sample size.

-Observational design: This study presents an observational analysis, with an eminently exploratory character, focused on identifying the potential of interprofessional education (IE) through clinical simulation to address stereotypes among health professionals, an area where research is still emerging. Although we recognize that a larger sample could provide more conclusive data, these initial results offer a basis for identifying patterns that can guide future studies with greater representativeness.

- Similar precedents. Similar studies, show that multiprofessional involvement is extremely difficult. Scotten, Mitzi et al (2015) analyze the impact of interprofessional training and attitudes of healthcare professionals, in relation to communication, and measured on the satisfactions of patients who were jointly cared for. This may indicate that the interest of the different professional groups in participating in non-technical skills development activities is low, and therefore the effort of researchers and teachers should be high in order to achieve high participation rates.

In our case, the measurement activity was carried out at the end of the 4-week academic activity, in May, which was scheduled last in terms of teaching planning for the master's degree in Medicine and Emergency Nursing. That is to say, the participating students began their master's academic training in September 2023 and finished this course in May 2024; as it was the last course of the course and the collection of information was voluntary, we understand that the response rate was low, due to the low motivation and reward, as well as the desire of the participating students to finish their studies.

Regarding the low proportion of participants from the medical community, similar studies show data on medical participation similar to ours, where the proportion of nurses is higher than that of physicians. He, Song et al. 2024.

COMMENTS: Thank you for revising the title, and providing reasons for the low response rate. Considering that this is the first study from the research group and is more of an exploratory pilot study nature, suggest changing the title to ‘Measuring Stereotypes in Interprofessional Education: A Pilot High-Fidelity Simulation Study Among Postgraduate Nursing and Physician Students in a Spanish University’. It would be better to specify that this is a study in a Spanish university, rather than ‘in Spain’, which might give the impression that this is a national study.

The introduction covers the definition and concept of IPE as well as teamwork extensively. The introduction can be condensed and focus on the key point i.e. stereotyping of health professionals to provide a clearer and more concise narrative.

RESPONSE

Thank you for your appreciation. Following this recommendation, we synthesized the text of the introduction, condensing information and thematic groups of evidence. We have reduced the length of the introduction by approximately 30%.

COMMENTS: Great to see that the introduction section has been consolidated. It seems to me that the length of the introduction has been reduced by more than half? I was wondering if more information on health professional identity and stereotyping could be provided, particularly on other similar studies published previously to justify the undertaking of this study.

Also in line 154, should be ‘professional affliations’. ‘s’ is missing.

It was stated that the objective of this study was to “identify the extent of ingrained stereotypical ideas and the impact of Interprofessional Training on the levels of knowledge, attitudes, and empathy towards other professionals among postgraduate nursing and medical students” . The levels of knowledge and empathy were not objectively assessed in this study hence this shouldn’t be included in the objective. In fact, there are two different objectives - 1. To identify the extent of ingrained stereotypical ideas in nursing and medical students; and 2. To understand the attitudes of nursing and medical students towards a high fidelity simulation in IPE.

For objective 2, the authors need to expand on how their high fidelity simulation in IPE study is situated in the body of literature – why is there a need for this work when there are many other articles on high fidelity simulation in IPE between nursing and medical students published previously?

RESPONSE

Thank you for your comment. Certainly, the interprofessional empathy and the attitudes towards the other group are not measured. We eliminated them as an objective, and focused on steotypes as the main object of study.
However, we would like to emphasize that professional empathy and attitudes are key to reducing negative interprofessional stereotypes. Empathy enables professionals to understand and value the experiences and contributions of colleagues from different disciplines, reducing preconceived judgments. This favors an open and collaborative attitude, essential to counteract stereotypes such as the perception of professional supremacy (as between doctors and nurses). Empathetic and respectful attitudes create a climate of cooperation, where stereotypical ideas about skills and roles are reduced, promoting positive and functional interprofessional relationships in the health team.
Therefore, we consider that the measurement of steotypes may reflect empathic attitudes towards the other profession. And interprofessional training programs could contribute to all these easily measurable constructs.

COMMENTS: The explanation above is great, could this be weaved into the introduction to justify the conduct of this study?

Duplicate references – 2 & 3, 73 & 74

RESPONSE

Indeed, these are duplicities; the typo is corrected. Thank you for your appreciation.

The study design needs to reconsidered as the findings from this study are based on a single ad hoc survey which has not undergone validity and reliability testings, and the conclusions were drawn using data from a small sample size. In-depth interviews may be better suited to meet the primary objective of this study, which is to assess the impact of stereotypes in postgraduate Nursing and Medical Students after completing interprofessional education.

RESPONSE

As previously mentioned, this is an observational and exploratory study. The qualitative design is taken into account as a next phase of this work, to be developed in the current academic year 204-25, with another group of students. Certainly, the qualitative approach could provide an additional focus that would help to shed light on different aspects of this field. This work serves as a starting point to initiate discussion topics.

COMMENTS:  Thanks for clarifying this. In this instance I would suggest emphasising that this is a pilot study in the title and throughout the manuscript, so that readers are aware of this. Doing so will also address the query on the small sample size to a certain extent.

The results section requires significant reworking. Tables 2-6 are not required and should be included in supplementary date. Presenting the normality analysis for each item in the main result section detract the readers from focusing on the main findings. Similarly, presenting the mean, median, mode etc for all items for a study with such a small sample size does not provide much value, and disrupts the flow of the presentation.

RESPONSE

Thank you for your comments. The results have been rewritten and the tables have been moved to the annexes section.

COMMENTS: Please update Table heading e.g. Table 7 should be Table 2.

Also, Table 2 should be revised to show the exact number for each response from disagree entirely to agree entirely. This is important as it looks like the number of responses received for each statement varies and also the sample size is relatively small, and so that the readers can see how the mean is derived if all of the numbers are being presented clearly across all 5 Likert-scale items. Also suggest to remove mean from the table as presenting mean for study with small sample sizes is misleading.

There is a potential for the survey to be used as an instrument to assess the impact of stereotypes in different groups of health professional students, however this will require a significant amount of further work to establish the validity

RESPONSE

Thank you for your consideration. As I have commented in previous answers, this is a first exploratory phase of further work; where the realization of focus groups is contemplated, from which, to elaborate questionnaires endowed with sufficient metric properties, with construct validity, content, measurement on a pilot group, with test-retest and finally a large multiprofessional sample. All this is part of a research project carried out within the group of researchers of the university, and in a doctoral program.

COMMENTS: Please see my previous comment on stating that this is a pilot study and future work has been planned to conduct follow-up studies.

Comments on the Quality of English Language

Proofreading is required.

Author Response

COMMENTS: Thank you for revising the title, and providing reasons for the low response rate. Considering that this is the first study from the research group and is more of an exploratory pilot study nature, suggest changing the title to ‘Measuring Stereotypes in Interprofessional Education: A Pilot High-Fidelity Simulation Study Among Postgraduate Nursing and Physician Students in a Spanish University’. It would be better to specify that this is a study in a Spanish university, rather than ‘in Spain’, which might give the impression that this is a national study.

RESPONSE.

Thank you for your feedback. With your permission, we find your suggested title highly appropriate and gladly accept it as such.

COMMENTS: Great to see that the introduction section has been consolidated. It seems to me that the length of the introduction has been reduced by more than half? I was wondering if more information on health professional identity and stereotyping could be provided, particularly on other similar studies published previously to justify the undertaking of this study.

Also in line 154, should be ‘professional affliations’. ‘s’ is missing.

RESPONSE

Thank you for your feedback. We will provide concise notes addressing professional stereotypes, particularly those related to nurses and physicians. Five references and two synthesized paragraphs have been added to meet your request while adhering to text length constraints. Additionally, the erroneous "s" that appeared due to a typographical mistake has been corrected.

COMMENTS: The explanation above is great, could this be weaved into the introduction to justify the conduct of this study?

RESPONSE

Thank you for your words. We added two paragraphs to the introduction, containing the message conveyed.

COMMENTS: Thanks for clarifying this. In this instance I would suggest emphasising that this is a pilot study in the title and throughout the manuscript, so that readers are aware of this. Doing so will also address the query on the small sample size to a certain extent.

RESPONSE

Thanks for your coments. The title has already been rectified and the appropriate rectifications have been included in section 2.1 study design and in the abstract.

COMMENTS: Please update Table heading e.g. Table 7 should be Table 2.

Also, Table 2 should be revised to show the exact number for each response from disagree entirely to agree entirely. This is important as it looks like the number of responses received for each statement varies and also the sample size is relatively small, and so that the readers can see how the mean is derived if all of the numbers are being presented clearly across all 5 Likert-scale items. Also suggest to remove mean from the table as presenting mean for study with small sample sizes is misleading.

RESPONSE

Thank you for your appreciation, but we consider maintaining the title of the tables, because although they are not provided in the main text of the article, they are maintained as supplementary material and can be consulted by the reader, if deemed necessary. Likewise, in the text the order in the narrative of the tables is maintained.

We think that changing the title of table 7 to table 2 could cause confusion for the reader. Therefore, we humbly ask to maintain the structure of table titles.

Following their recommendations, Table 7 includes the distribution of all the responses given, while eliminating the mean and standard deviation as data..

COMMENTS: Please see my previous comment on stating that this is a pilot study and future work has been planned to conduct follow-up studies.

RESPONSE

Thanks for your appreciations. They have incorporated the clarification as a pilot study in the title, as previously suggested.

Reviewer 4 Report

Comments and Suggestions for Authors

I have no more comments.

Author Response

Thank you very much for your corrections.

In the version that we attach now, we incorporate the corrections of one of the reviewers.
